# Novel *FANCI* and *RAD54B* Variants and the Observed Clinical Outcomes in a Hungarian Melanoma Cohort

**DOI:** 10.3390/ijms26010023

**Published:** 2024-12-24

**Authors:** Barbara Anna Bokor, Aliasgari Abdolreza, Flóra Kaptás, Margit Pál, Zita Battyani, Márta Széll, Nikoletta Nagy

**Affiliations:** 1Department of Medical Genetics, University of Szeged, 6720 Szeged, Hungary; bokor.barbara.anna@med.u-szeged.hu (B.A.B.); aliasgari.abdolreza@gmail.com (A.A.); kaptasflora@gmail.com (F.K.); pal.margit@med.u-szeged.hu (M.P.); szell.marta@med.u-szeged.hu (M.S.); 2HUN-REN-SZTE Functional Clinical Genetics Research Group, University of Szeged, 6720 Szeged, Hungary; 3Mór Kaposi Teaching Hospital, 7400 Kaposvár, Hungary; battyani_z@yahoo.com

**Keywords:** melanoma, germline, progression, therapy resistance, patients’ survival, likely pathogenic variant, novel variant

## Abstract

Accumulating evidence suggests that inherited melanoma is not rare and approx. one in seven individuals with melanoma has clinically relevant hereditable cancer-predisposing and/or -susceptibility variant(s). Concerning its germline genetic background, genetic screening aims to identify either variants of predisposing genes with high penetrance or variants of susceptibility genes with medium or low penetrance. However, less attention is paid to genetic testing of germline variants of genes influencing patients’ survival outcomes or enhancing the design of new therapies. We aimed to investigate whether the germline genetic background of a Hungarian melanoma cohort (*n* = 17) contains any pathogenic or likely pathogenic variants of the *BRCA2*, *POLE*, *WRN*, *FANCI*, *PALB2*, and *RAD54B* genes and if the presence of these variants correlate with the clinical findings of the patients, including the advanced stage of melanoma, poor prognosis, and poor survival. We identified three novel variants in the *FANCI* gene and one novel variant in the *RAD54B* gene. We detected rapid disease progression, unfavorable outcome, and therapeutic resistance in the patient carrying the likely pathogenic *FANCI* variant. Our study highlights the importance of screening germline variants of genes influencing melanoma progression, therapy resistance, and survival of patients.

## 1. Introduction

Malignant melanoma is recognized as a complex disease, its development being influenced by genetic, environmental, and lifestyle factors [1,2,3]. Accumulating evidence suggests that inherited melanoma is not rare and approx. one in seven individuals with melanoma has clinically relevant hereditable cancer-predisposing or -susceptibility variant(s) [4]. Concerning the germline genetic background of melanoma, genetic screening aims to identify either variants of predisposing genes with high penetrance (*CDKN2A*, *CDK4*, *BAP1*, *POT1*, *ACD*, *TERF2IP*, and *TERT*) or variants of susceptibility genes with medium or low penetrance (*MC1R*, *MITF*, *SLC45A2*, *TYR*, *OCA2*, *ASIP*, *PL2G6*, *FTO*, *PARP1*, *ATM*, *CDKAL1*, *CCND1*, and *CYP1B1*), which are known to play a major role in the genetic background of melanoma [1,5].

We have recently reported a Hungarian melanoma cohort (*n* = 17) with increased risk [5]. All of them had at least three dysplastic naevi diagnosed by expert dermatologists and proved by dermatohistological examinations [5]. Fourteen patients were diagnosed with malignant melanoma and three patients had dysplastic naevus syndrome [5]. Using a gene panel of the melanoma-predisposing and melanoma-susceptibility genes described above, germline genetic variants of genes were identified in 10 of the 17 patients (58.82%) [5].

In addition to the germline variants of the melanoma-predisposing and melanoma-susceptibility genes, the accumulating evidence suggests that germline variants of genes, involved in DNS repair mechanisms, have been implicated in rendering melanoma patients more susceptible to tumor progression and affecting their response to treatments [6]. Here, our aim was to investigate whether patients in the Hungarian melanoma cohort (*n* = 17) with increased risk carry any pathogenic or likely pathogenic germline variants of the *BRCA2*, *POLE*, *WRN*, *FANCI*, *PALB2*, and *RAD54B* genes associated with melanoma survival and response to therapy. We also investigated whether the presence of these variants correlates with the clinical findings of the patients, including the advanced stage of melanoma, poor prognosis, and poor survival.

## 2. Results

### 2.1. In Silico Variant Analysis

We identified mutations using a six-gene panel in four of the 17 patients (23.5%). None of them overlaps with the variants reported by Amaral et al. (2020) on the *BRCA2*, *POLE*, *WRN*, *FANCI*, *PALB2*, and *RAD54B* genes [6]. However, we identified three novel variants in the Fanconi anemia, complementation group I gene (*FANCI*) in three patients (patients 9, 15, and 16), and one novel variant in the RAD54 homolog B gene (*RAD54B*) in one melanoma patient (patient 14) (Figure 1).

The novel c.3111_3123del, p.Ser1038LeufsTer19 variant of the *FANCI* gene (15q26.1; NM_001113378.2) is a nonsense variant in exon 29 resulting in the formation of a premature termination codon after the 1038th amino acid of the polypeptide (Figure 2). Based on the ACMG classification guideline, this variant is classified as likely pathogenic, considering that this is a null variant in a gene where loss of function is a known mechanism of disease (PVS1) and has an extremely low frequency in the gnomAD database (PM2).

The novel c.2768A > G, p.Tyr923Cys variant of the *FANCI* gene (15q26.1; NM_001113378.2) is a missense variant in exon 25 causing a tyrosine-to-cysteine amino acid change in the 923th position of the protein (Figure 2). According to the ACMG classification guideline, this variant is classified as a variant of unknown significance, considering the extremely low frequency of the variant in the gnomAD population databases (PM2). EVE (evolutionary model of variant effect; https://evemodel.org/; accessed on 14 November 2024) suggests pathogenic effect (Figure 3a) and other in silico prediction tools also support a deleterious effect of the variant (MT, DANN, Canonym, fitCons), while others report an uncertain effect (REVEL, MUT Assessor, SIFT, FATHMM, BayesDel).

The c.3896G > T, p.Arg1299Leu variant of the *FANCI* gene (15q26.1; NM_001113378.2) is a missense variant in exon 37 causing an arginine-to-leucine amino acid change at the 1299th position of the protein (Figure 2). According to the ACMG classification guideline, this variant is classified as a variant of unknown significance, considering the extremely low frequency of the variant in the gnomAD population databases (PM2). Some of the in silico prediction tools support a deleterious effect of the variant (MT, DANN, GenoCanyon, fitCons), while other tools such as EVE (Figure 3b), SIFT, FATHMM, and MetaLR predict an uncertain effect. This variant was previously only published in one paper as a candidate for susceptibility to ovarian cancer [7].

The p.Tyr923Cys variant affects the functional domain FANCI solenoid 3 (position 787–972 amino acids) and the variant p.Ser1038LeufsTer19 affects the functional domain FANCI solenoid 4 (position 985–1236 amino acids). The p.Arg1299Leu variant does not affect any known functional domain of the FANCI protein (SMART Protein, https://smart.embl.de/smart/show_motifs.pl?ID=Q9NVI1-1&DO_PFAM=DO_PFAM&; accessed on 14 November 2024) (Figure 2).

The novel c.337A > G p.Lys113Glu variant of the *RAD54B* gene (8q22.1; NM_012415.3) is a missense variant in exon 4 causing a lysine-to-glutamine amino acid change in the 113th position of the protein (Figure 2). Based on the ACMG classification guideline, this variant is classified as a variant of unknown significance, considering the extremely low frequency of the variant in the gnomAD population databases (PM2), and also the fact that in silico prediction tools unanimously support a benign effect on the gene (BP4) (Revel, MUT Assessor, MT, PrimateAI, BayesDel, SpliceAI).

The identified variants do not affect any non-coding RNA regions (Ensemble Genome Browser; Ensembl release 113, October 2024).

### 2.2. Clinical Outcomes

Amaral et al. (2020) identified an association between the reported variants of the *BRCA2*, *POLE*, *WRN*, *FANCI*, *PALB2*, and *RAD54B* genes and the patients’ clinical outcomes as well as the therapy resistance [6]. Therefore, in the case of the four variants (three on the *FANCI* gene and one on the *RAD54B* gene) identified by this study, we also analyzed the clinical characteristics of the patients who harbor these variants.

The likely pathogenic variant in the *FANCI* gene is present in patient 15, who was first diagnosed with melanoma malignum at the age of 31 years (1999) [5]. The staging examinations showed no signs of metastases; consequently, only the excision of the melanoma was performed, without any additional treatment. After treatment for malignant melanoma, the patient attended follow-up examinations yearly, which did not show signs of late metastases, relapse, or second primary melanomas in the following years. In March 2023, a neck, thorax, abdominal, and pelvic CT scan was performed on the asymptomatic patient, where suspicion of multiplex cerebral and lung metastases, multiplex metastases in the liver and spleen, as well as lymphadenomegalia colli l.s. et hilii, and contrast accumulation in the gall bladder and bladder were reported. Contrast cranial MRI confirmed the presence of multiplex brain metastases. The tissue biopsy taken from the liver and subsequent histopathology showed a liver metastasis of melanoma malignum. A careful clinical examination did not show any signs of the primary tumor, but the presence of a second primary melanoma malignum was suspected behind the disseminated multiplex metastases.

In patient 15, a genetic examination of the tissue biopsy showed BRAF positivity; however, considering the advanced and disseminated nature of the disease, especially the presence of brain metastases, palliative whole-brain radiation therapy (WBRT) was performed and a request for ipilimumab + nivolumab immunotherapy was submitted. Meanwhile, dabrafenib + trametinib targeted molecular therapy was administered for 3 months between April and July 2023, leading to a clinical improvement in the patient’s condition. In July 2023, ipilimumab + nivolumab therapy was initiated and after three months it was switched to nivolumab monotherapy. With the continuous administration of ipilimumab + nivolumab combined immunoterapy, and then nivolumab monotherapy, the condition of the patient remained stable until the end of November 2023, when rapid clinical progression was observed, indicating resistance to immunotherapy. The treatment of the patient was once again switched to dabrafenib + trametinib combined molecular targeted therapy, but the therapy could not control the further progression of the disease leading to the patient’s exit in March 2024.

The advanced nature of metastatic disease at the time of diagnosis in November 2023, the unfavorable prognosis of the disease, and resistance to immunotherapy and targeted molecular therapy in the presence of a likely pathogenic *FANCI* variant in the patient support the possible disease-modifying role of the *FANCI* gene in patients with malignant melanoma (Table 1).

In the case of patient 9, we identified another novel *FANCI* variant, the c.2768A > G, p.Tyr923Cys. The female patient was diagnosed with melanoma malignum at the age of 42 years and in the absence of lymphatic or other metastases, only an excision of the melanoma was performed in 2020, without adjuvant therapy. The patient had a positive family history of malignant melanoma (aunt on the father’s side). Patient 9 is under regular dermatological care, and her condition is unchanged as of December 2024.

We also identified the c.3896G > T, p.Arg1299Leu variant in the *FANCI* gene in a 53-year-old female patient with dysplastic naevus syndrome, who had multiple dysplastic naevi removed, but in her case no melanoma malignum was observed yet. She also had a positive family history of melanoma malignum, as her father was affected by the disease. The patient is under regular dermatological care and her condition is unchanged as of December 2024.

In the case of the two novel VUS variants we identified in the *FANCI* gene, we were unable to establish any disease-modifying role based on the available clinical data of our patients, so further studies and careful follow-up of these patients are needed to determine their role in melanoma disease progression and therapeutic response.

Additionally, we identified a VUS variant in *RAD54B* in a 43-year-old female patient (patient 14), who had a stage-pT4 melanoma malignum at the time of diagnosis, without lymphatic involvement or other metastases. After excision of the cutaneous melanoma in 2019, no other therapy was administered, and after 5 years of follow-up the patient remains in remission. Based on this, we could not identify any evidence supporting the disease-modifying role of the germline *RAD54B* c.337A > G p.Lys113Glu variant regarding the unfavorable outcome, progression of malignant melanoma, or resistance to immunotherapy. However, the fact that she carries a VUS variant in a gene (RAD54B) that has been implicated in disease progression alerts us that she needs careful follow-up.

## 3. Discussion

Here, we report the genetic examination of a Hungarian melanoma cohort with increased risk. In our previous publication, we have summarized the germline variants of melanoma-predisposing and melanoma-susceptibility genes [5]. However, less attention is paid to genetic testing of germline variants of genes influencing patients’ survival outcomes or enhancing the design of new therapies [6]. Here, we investigated whether melanoma patients in this published cohort harbor pathogenic or likely pathogenic germline variants in genes associated with unfavorable clinical outcomes [6].

The germline variants of *BRCA2*, *POLE*, *WRN*, *FANCI*, *PALB2*, and *RAD54B* genes, involved in DNS repair mechanisms, have been implicated in rendering melanoma patients more susceptible to tumor progression and affecting their response to treatments [8]. BRCA2 protein is involved in maintenance of genome stability, specifically the homologous recombination pathway for double-strand DNA repair. In *BRCA2* mutation carriers, both uveal melanoma and cutaneous melanoma were found at significantly increased frequency [9,10,11]. Additionally, the germline variants in *BRCA2* have been found to increase the risk of melanoma and affect survival rates [12]. POLE and WRN are involved in maintaining genomic integrity through DNA replication and repair. Germline variants in these genes may impair these functions, contributing to higher levels of genomic instability in melanoma cells [13,14,15].

Additionally, variants of *FANCI*, *PALB2*, and *RAD54B* are associated with altered survival outcomes in melanoma patients [8,16,17]. *FANCI* is part of the Fanconi anemia pathway, which is vital for interstrand cross-link repair. Variants of *FANCI* may enhance DNA damage accumulation in melanoma cells, which may promote more aggressive cancer characteristics [17,18]. PALB2 protein partners with BRCA2 in homologous recombination, and mutations in *PALB2* are similarly implicated in an increased melanoma risk and poorer survival [7].

FANCI has four distinct alpha solenoid segments (S1–S4). Regarding the three novel *FANCI* variants identified by this study, the p.Ser1038LeufsTer19 variant affects the solenoid 4 domain, and the p.Tyr923Cys variant is located within the solenoid 3 domain on the FANCI protein (Figure 3). Our results correlate well with previous findings, as heterozygous germline deletion in exon 9 reported by Amaral et al. (2020) is also located within the solenoid 3 domain of the FANCI protein [6].

Accumulating evidence suggests that genes associated with oncogenic pathways are identified as potential mini-drivers in tumor development [19]. Patients with rare pathogenic or likely pathogenic variants in mini-driver genes are association with worse tumor prognosis [19]. Further studies are needed to investigate the putative mini-drivers in melanoma.

In accordance with the observations about germline pathogenic and likely pathogenic *FANCI* variants in the literature, in the case of the novel, likely pathogenic c.3111_3123del, p.Ser1038LeufsTer19 variant in the *FANCI* gene, we detected a strong link with rapid disease progression, unfavorable outcome, and therapeutic resistance based on available clinical and genetic data of the patient. In the case of the other VUS variants in the *FANCI* and *RAD54B* genes in our Hungarian cohort, we observed no effect on disease progression or therapeutic response. Further studies are needed to report and highlight the clinical importance and relevance of genetic screening of putative germline variants that influence disease progression or therapeutic response in patients with melanoma.

## 4. Materials and Methods

### 4.1. Patients

In our current analysis, we included 17 Hungarian, unrelated melanoma patients, 10 females and 7 males. 14 patients were diagnosed with melanoma malignum, while 3 patients were diagnosed with dysplastic naevus syndrome. The histological characteristics of the tumors are summarized in Table 2.

The mean age of the melanoma patients at the time of diagnosis of the first melanoma malignum was 49.5 years. Family history, the presence of any other immune system disease, and the detected *CDKN2A* variants are summarized in Table 3.

Further detailed clinical characteristics of the 17 members of the Hungarian malignant melanoma cohort with increased risk are summarized in our previous publication [5]. After genetic counseling and obtaining the written informed consent of the enrolled individuals, peripheral blood samples were taken, and genomic DNA was isolated using the QIAGEN DNeasy kit (Qiagen, Hilden, Northrine-Westfalia, Germany).

Of the 17 patients, 3 patients had more than one primary melanoma (patients 5, 6, and 15). Two patients had lymphatic metastases (patients 6 and 15), and only one patient (patient 15) had additional multiple metastases (multiplex brain, spleen, lung, liver metastases, contrast accumulation in the gallbladder and bladder). Each patient diagnosed with melanoma malignum underwent excision of the melanotic lesion, three patients received immunotherapy (patients 6, 12, and 15), one patient received combined targeted molecular therapy (patient 14), while none of the patients required traditional chemotherapy. Patient 15 underwent palliative radiotherapy for brain metastases (whole-brain radiation therapy, WBRT) (Table 1).

### 4.2. Targeted Next-Generation Sequencing with a Virtual Gene Panel

Patients’ genotypes were determined using a targeted next-generation sequencing (NGS) approach. Libraries were prepared using the SureSelectQXT Reagent Kit (Agilent Technologies, Santa Clara, CA, USA). Pooled libraries were sequenced on the Illumina NextSeq 550 NGS platform using the 300-cycle Mid Output Kit v2.5 (Illumina, Inc., San Diego, CA, USA). Adapter-trimmed and Q30-filtered paired-end reads were aligned to the hg19 human reference genome using the Burrows–Wheeler Aligner (BWA). Duplicates were marked using the Picard software package. The Genome Analysis Toolkit (GATK) was used for variant calling (BaseSpace BWA Enrichment Workflow v2.1.1. with BWA 0.7.7-isis-1.0.0, Picard: 1.79, and GATK v1.6-23-gf0210b3).

Sequencing revealed that the mean on-target coverage was 71× per base with an average percentage of targets covered greater than or equal to 30×, respectively. Variants passed through the GATK filter were used for downstream analysis and annotated using the ANNOVAR software tool (version 17 July 2017). Single-nucleotide polymorphism testing was performed as follows: high-quality sequences were aligned with the human reference genome (GRCh38/hg19) to detect sequence variants, which were analyzed and annotated. Variants were filtered according to read depth, allele frequency, and prevalence reported in genomic variant databases, such as ExAc (v.0.3) and Kaviar. Variant prioritization tools (PolyPhen-2, SIFT, LRT, Mutation Assessor) were used to predict the functional impact of the mutation. We interpreted the sequencing results using the Franklin Genoox website, which creates and uses a virtual panel that includes 6 genes (*BRCA2*, *POLE*, *WRN*, *FANCI*, *PALB2*, and *RAD54B*) influencing melanoma prognosis and survival [6].

The candidate variants were confirmed by bidirectional capillary Sanger sequencing carried out according to the standard protocol with an Applied Biosystems 3500 Genetic Analyzer (Thermo Fisher Scientific, Waltham, MA, USA). Regions of 500 nucleotides upstream and downstream from the identified variants were picked from the Ensemble Genome Browser (https://www.ensembl.org/index.html?redirect=no; accessed on 3 July 2024) and used for the designing of the primers on the online website Blast Primer Designer (https://www.ncbi.nlm.nih.gov/tools/primer-blast/; accessed on 3 July 2024). The sequences of the primers used for Sanger sequencing are listed in Table 4.

Information regarding non-coding elements was searched by the Ensemble Genome Browser.

## Figures and Tables

**Figure 1 ijms-26-00023-f001:**
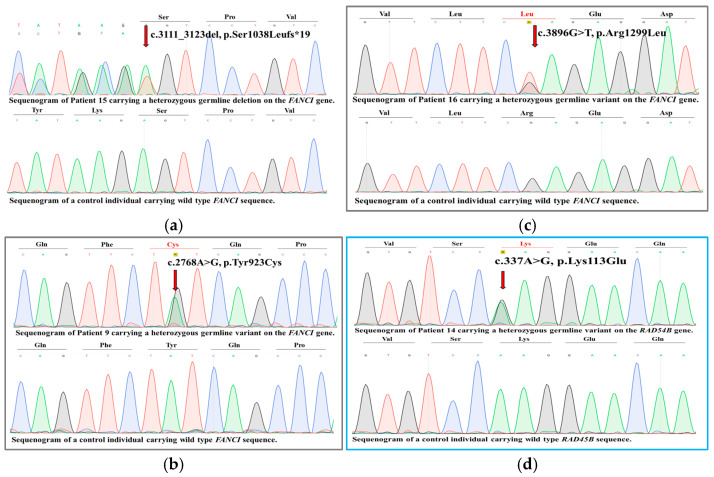
Sequenograms of the novel *FANCI* and *RAD54B* variants identified in the Hungarian melanoma cohort (*n* = 17). (**a**) The novel likely pathogenic *FANCI* variant c.3111_3123del, p.Ser1038LeufsTer19 is carried by patient 15. (**b**) Among the novel missense VUS FANCI variants, the c.2768A > G, p.Tyr923Cys is present in patient 9, and (**c**) the c.3896G > T, p.Arg1299Leu is present in patient 16. (**d**) The novel likely pathogenic *RAD54B* variant is detected in patient 14 (*FANCI* sequenograms are surrounded by grey, while *RAD54B* sequenograms are surrounded by light blue frames).

**Figure 2 ijms-26-00023-f002:**
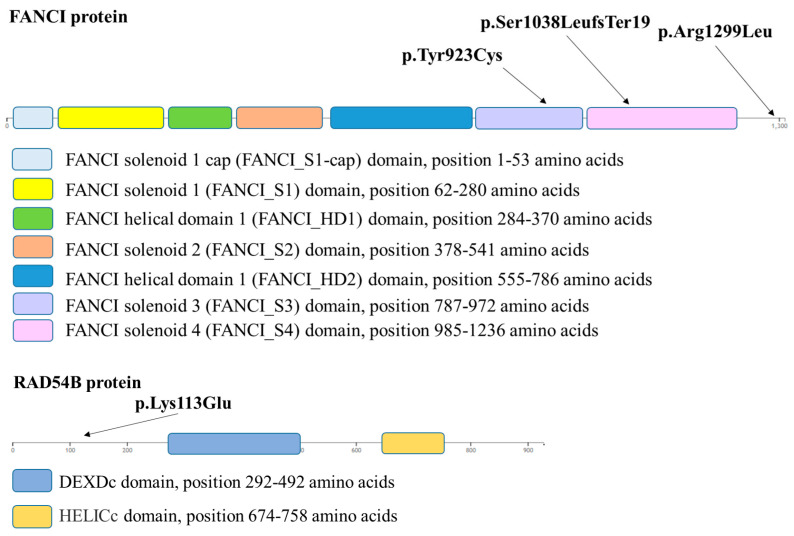
Position of the identified variants on the FANCI and on the RAD54B proteins (https://www.rcsb.org/sequence/3s51; accessed on 14 November 2024).

**Figure 3 ijms-26-00023-f003:**
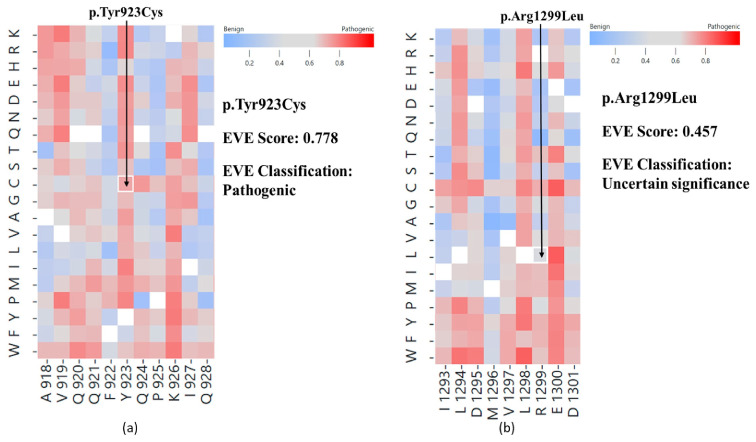
Heat maps of the evolutionary model of variant effect (EVE) scores of the missense variants of the FANCI protein. (**a**) Regarding the high EVE score (0.778) of the p.Tyr923Cys missense variant, it is classified as pathogenic according to the EVE classification. (**b**) Based on the medium EVE score (0.457) of the p.Arg1299Leu missense variant, it is classified as uncertain significance according to the EVE classification (https://evemodel.org/; accessed on 14 November 2024).

**Table 1 ijms-26-00023-t001:** Clinical characteristics and therapies administered in the Hungarian cohort (*n* = 17). Patients carrying either *FANCI* or *RAD54B* variants are highlighted with a gray background.

Patient No.	Age of Onset	No. of Primary Melanomas	Lymphatic Metastasis	Other Metastasis	Therapy				
					Excision	Targeted Molecular Therapy	Immuno- Therapy	Radio- Therapy	Chemo-Therapy
1	58	1	no	no	yes	no	no	no	no
2	76	1	no	no	yes	no	no	no	no
3	82	1	no	no	yes	no	no	no	no
4	55	1	no	no	yes	no	no	no	no
5	26	2	no	no	yes	no	no	no	no
6	51	3	yes	no	yes	no	yes	no	no
7	44	0	no	no	no	no	no	no	no
8	57	0	no	no	no	no	no	no	no
9	42	1	no	no	yes	no	no	no	no
10	42	1	no	no	yes	no	no	no	no
11	51	1	no	no	yes	no	no	no	no
12	50	1	no	no	yes	no	yes	no	no
13	42	1	no	no	yes	no	no	no	no
14	43	1	no	no	yes	no	no	no	no
15	31	1	yes	yes, multiple	yes	yes	yes	yes	no
16	53	0	no	no	no	no	no	no	no
17	40	1	no	no	yes	no	no	no	no

**Table 2 ijms-26-00023-t002:** Histological characteristics of the melanoma in the Hungarian cohort (*n* = 17).

Patient No.	pTNM Stage	Breslow Thickness (mm)	Clark Level	Mitotic Rate (mm^2^)	Presence of Ulceration/ Regression
1	pT2a	1.28	III	1–26	no
2	pT1a	0.5	II	0	regression
3	NA	NA	NA	NA	NA
4	pT4a	6	IV	2	ulceration
5	pT1a	0.35	II	2	no
	pT1a	0.32	II	1	no
6	pT3a	3.15	IV	2–3	regression
	pT2a	2	III	2–3	regression
	in situ superficially spreading melanoma				
7	patient is affected by dyspastic naevus syndrome				
8	patient is affected by dysplastic naevus syndrome				
9	pT3a	0.78	IV	2–3	ulceration
10	pT1a	0.75	III	2	no
11	pT1b	0.6	IV	2	ulceration
12	pT2a	1.9	IV	1	no
13	pT1a	0.58	III	0	no
14	pT4a	4.7	III	3–4	no
15	pT1a	0.8	II	0	no
16	patient is affected by dyspastic naevus syndrome				
17	pT2a	1.21	IV	2	no

**Table 3 ijms-26-00023-t003:** Clinical features of and the presence of any *CDKN2A* variants in the patients (*n* = 17).

Patient No.	Immune System Disease	Family History of Melanoma	Presence of Any Germline *CDKN2A* Variant
1	unknown	negative	no
2	unknown	positive	no
3	unknown	negative	no
4	unknown	negative	yes, a leaning-pathogenic VUS
5	unknown	negative	no
6	unknown	negative	no
7	unknown	negative	no
8	ulcerative colitis	negative	no
9	unknown	positive	no
10	unknown	negative	no
11	unknown	negative	no
12	unknown	negative	no
13	unknown	negative	no
14	unknown	negative	no
15	unknown	negative	no
16	unknown	positive	no
17	unknown	negative	no

**Table 4 ijms-26-00023-t004:** Primers used for the confirmation of the candidate variants with Sanger sequencing.

*FANCI* exon 25 forward primer	TTGTGGGGAGATTACACAACC
*FANCI* exon 25 reverse primer	TCTCAAGTGTCTTCTGGTAGGT
*FANCI* exon 25 forward primer	CAATACCACTTTCTCCTGCTTC
*FANCI* exon 25 reverse primer	CAGCCACTCTTTGTGGTTGA
*FANCI* exon 37 forward primer	GTGCGTGCTTGCTTTAGGTA
*FANCI* exon 37 reverse primer	ATCAAACAAGTCGGGGCAAC
*RAD54B* exon 4 forward primer	TGTGCCTTTTGGTTTTGTTTGAAT
*RAD54B* exon 4 reverse primer	AGATTGTCAGGCTCACTAACCA

## Data Availability

The data presented in this study are available on request from the corresponding authors. The data are not publicly available because they are genetic data.

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
