# Peer review of "Novel FANCI and RAD54B Variants and the Observed Clinical Outcomes in a Hungarian Melanoma Cohort"

_ijms, 2024, doi:10.3390/ijms26010023_

Round 1
Reviewer 1 Report
Comments and Suggestions for Authors
Well-written, well-presented manuscript with important therapeutical consequences. In line 141 please correct melanoma maignum to melanoma malignum.
There is a promising progress in the therapy of malignant melanoma in the last decade. However, the histological investigation of the tumors give us a lot of information, and we can differentiate low risk and high risk tumors, in many cases it is hard to predict the exact clinical outcome of the disease. Detection of new mutations could provide new information about the therapeutic response of the patients, for this, there is a great interest of the molecular background and the genetic investigations of melanoma. This manuscript presents 3 novel variants of FANCI gene, and one novel mutation of RAD54B gene, these mutations were newly recognized in melanoma patients. In the first patient (Patient 15) it seems like the new clinical variant is in association with an unfavourable outcome with a late relapse and progression of the tumor. In the other two patients, the late clinical outcome is not known yet. The RAD54B variants probably showed a favourable outcome in a Patient with a high risk tumor.
1. 1. It would be clear if the histological parameters of the tumor were also signed in the manuscript (Patient 15, 9 and 14: pTNM stage, Breslow thickness, Clark level, mitotic rate, presence of ulceration or regression).
2. 2. Regarding to the Question 1, how can they define „increased risk” in melanoma or dysplastic naevus syndrome patients?
3. 3. How did they classify the genetic variants? What was the basis of the classification?
References are appropiate and cited from the recent years. The topic is relevant, the conlcusions are consistent with the evidences, although the cohort is small.
In summary, interesting results in a current topic with promising clinical impact.
Author Response
"Please see the attachment."

Reviewer 2 Report
Comments and Suggestions for Authors
Comments:
1. Figure 3: where is p-Lys113Glu? please label it.
2. Table 1: More patients' information needed: BMI/BRI, alcohol and smoking history, family history of melanoma, any other immune system diseases, CDKN2A mutation, etc.
3. All patients have same race? skin color?
4. Table 1: please separate FANCI and RAD54B in different color.
5. Any similar data and report from other countries? please discuss.
Author Response
"Please see the attachment."

Reviewer 3 Report
Comments and Suggestions for Authors
Dear authors,
Your manuscript "Novel FANCI and RAD54B variants and the observed clinical outcomes in a Hungarian melanoma cohort" presents the results of the genetic profiling of 17 Hungarian patients with melanoma, reporting variants with no previous description in the cancer context. These variants were screened by NGS and confirmed by Sanger sequencing. Clinically, you reported that a novel mutation in the FANCI gene could be responsible for treatment resistance and other worsening factors in the patient. Despite the potential contribution of this study to their field, I would like to comment on some concerns.
Major comments
1. Major findings are described in a small and heterogeneous sample with 17 individuals. Could you describe the statistical power of this sample, please? How representative is it?
2. Please describe all the methods in detail, they must be reproducible. For example, you wrote, "genomic DNA was isolated using QIAGEN kits.". Which kits were used? In addition, describe Sanger sequencing methods. Which primers were used? How did you choose the regions to be analyzed?
3. In terms of the biology of potential mechanisms driven by germline secondary mutations, the perspective of this paper could fit with the mini-driver hypothesis. It intends to explain how secondary contributions could worsen some cancers' prognoses. A recent study has figured out how mutations in POLE are associated with alterations in the expression of other genes (https://peerj.com/articles/15410/). As this gene was also relevant to your study, I suggest exploring this concept more in the context of your research and discussing these results.
4. Following the previous comment, it would be relevant to know if the mutational sites described in the paper include any non-coding RNA region (miRNA, lncRNA, etc). Maybe, no canonical ways could support your results and conclusions.
Author Response
"Please see the attachment."

Round 2
Reviewer 2 Report
Comments and Suggestions for Authors
No more comments.
Reviewer 3 Report
Comments and Suggestions for Authors
Dear authors,
Your manuscript "Novel FANCI and RAD54B variants and the observed clinical outcomes in a Hungarian melanoma cohort" presents the results of the genetic profiling of 17 Hungarian patients with melanoma, reporting variants with no previous description in the cancer context. These variants were screened by NGS and confirmed by Sanger sequencing. Clinically, you reported that a novel mutation in the FANCI gene could be responsible for treatment resistance and other worsening factors in the patient. Thank you for having responded to my previous comments.